# Boosting Semi-Supervised Few-Shot Object Detection with SoftER Teacher

## Abstract

Few-shot object detection (FSOD) is an emerging problem aimed at detecting novel concepts from few exemplars. Existing approaches to FSOD assume abundant base labels to adapt to novel objects. This paper studies the task of *semi-supervised FSOD* by considering a realistic scenario in which both base and novel labels are simultaneously scarce. We explore the utility of unlabeled data and discover its remarkable ability to boost semi-supervised FSOD by way of region proposals. Motivated by this finding, we introduce SoftER Teacher, a robust detector combining pseudo-labeling with representation learning on region proposals, to harness unlabeled data for improved FSOD without relying on abundant labels. Extensive experiments show that SoftER Teacher surpasses the novel performance of a strong supervised detector using only 10% of required base labels, without experiencing catastrophic forgetting observed in prior approaches. Our work also sheds light on a potential relationship between semi-supervised and few-shot detection suggesting that a stronger semi-supervised detector leads to a more effective few-shot detector.

## 1 Introduction

Modern object detection systems have enjoyed tremendous progress in recent years, with many successful applications across diverse industries. Their success can be mainly attributed to the availability of large-scale, well-annotated datasets such as the MS-COCO benchmark [28]. However, the demand for more powerful and accurate detection models requires considerable investments in the hand-labeling of massive amounts of data, which are costly to scale. Thus, there is a growing trend in the community to shift toward a more *label-efficient* paradigm, one that can enable detection models to do more with less hand-labeled data. Such emerging research directions include self-supervised representation learning from unlabeled data [2, 7, 39], multi-modal pre-training from Web-labeled data [25, 32], semi-supervised detection (SSOD) [30, 48, 52], and few-shot detection (FSOD) [10, 11, 47], all of which have shown great promise in alleviating the dependency on large amounts of instance-level class annotations and bounding boxes.

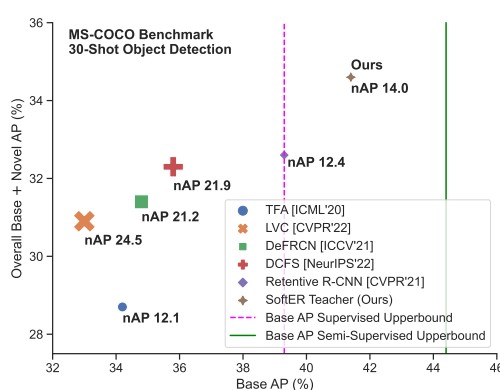

**Figure 1:** The evaluation of *generalized* FSOD is characterized by the trade-off between novel accuracy and base forgetting. We leverage unlabeled data to optimize for semi-supervised FSOD on both classes. Our approach significantly expands base class AP ($39.3 \rightarrow 44.4$) while exhibiting less than 7% in base degradation (*vs.* 16% for LVC [21]). SoftER Teacher is the best model on the `Overall` AP metric, leading the next best Retentive R-CNN [10] by $+2.0$ AP.

Submitted to 37th Conference on Neural Information Processing Systems (NeurIPS 2023). Do not distribute.

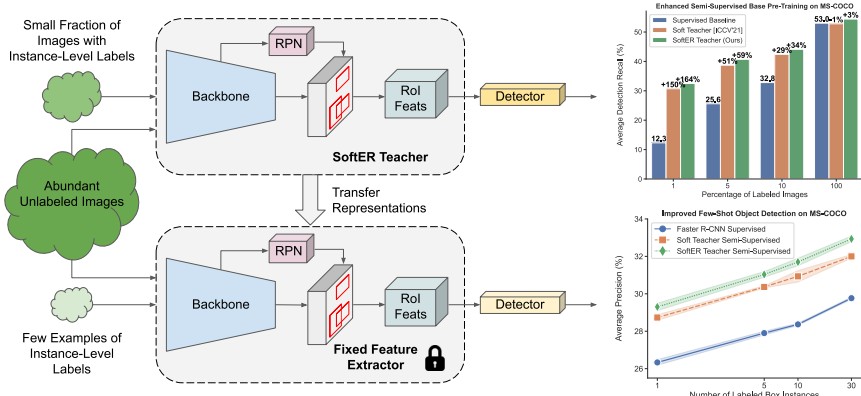

**Figure 2:** We present the Label-Efficient Detection framework to harness supplementary unlabeled data for generalized semi-supervised few-shot detection. At the core of the framework is our proposed SoftER Teacher with Entropy Regression for improved semi-supervised base representation learning (**upper right**). Extensive comparative experiments show that SoftER Teacher is also a more label-efficient few-shot detector (**lower right**).

This paper focuses on the intersection of SSOD and FSOD, which are essentially two sides of the same coin in the context of label-efficient detection. On one side, SSOD investigates the detection problem with a *small fraction of images* containing ground truth labels. On the other side, FSOD addresses the objective of adapting a *base* detector to learn *novel* concepts from *few instance-level* annotations. Existing approaches to FSOD assume abundant base classes to train the base detector. However, such assumption is not ideal in practical scenarios where labels may be limited for both base and novel classes, giving rise to the research question: ***can we do more on FSOD with the available unlabeled data without additional hand-labeling?***

We answer this question by introducing the unique task of *semi-supervised few-shot detection*, in which we explore the utility of unlabeled data for improving detection with label scarcity for both base and novel classes. Inspired by recent advances in SSOD and FSOD, our approach is two-fold: (1) we leverage unlabeled data to improve detection with a small fraction of labeled images; and (2) we generalize the resulting semi-supervised detector into a label-efficient few-shot detector by way of transfer learning. Our chief motivation is to not necessarily depend on an abundance of base classes for robust few-shot detection, which increases the versatility of our approach in realistic applications.

Moreover, our approach to semi-supervised FSOD adapts a base detector to learn novel concepts with *reduced performance degradation to base classes*, a desirable result missing in most prior approaches. Figure 1 illustrates that while recent work [11, 21, 36] achieve impressive detection on novel categories, *they all ignore the importance of preserving base class accuracy*. For *generalized* FSOD [10], the goal is to expand the learned vocabulary of the base detector with novel concepts. As such, base and novel class performances are equally important, since samples at test time may contain instances of both objects. Therefore, the more realistic evaluation metric for FSOD is not only novel AP, but the combined base and novel AP, for which our approach establishes a new state of the art.

We measure the utility of unlabeled data within our integrated semi-supervised few-shot framework, and discover an interesting empirical finding connecting the effectiveness of unlabeled data to semi-supervised FSOD by way of region proposals. Without bells and whistles, by simply adding unlabeled data to a supervised detector, we show a marked improvement on both base and novel class performances while also mitigating catastrophic base forgetting [31].

**Summary of Main Contributions.**    First, we introduce SoftER Teacher, a simple yet effective and versatile detector, to combine the strengths of pseudo-labeling with representation learning on unlabeled images. SoftER Teacher enhances the quality of region proposals to substantially boost semi-supervised FSOD. Our empirical analysis on the relationship between unlabeled data and region proposals extends earlier results on proposal evaluation beyond supervised detection [17, 46].

Second, Figure 2 illustrates a potential relationship suggesting that a strong semi-supervised detector is also a label-efficient few-shot detector, an interesting and non-trivial empirical observation linking the two disparate domains. On the task of semi-supervised FSOD, our SoftER Teacher model exceeds the novel class performance of a strong supervised detector [10] using only 10% of required base

labels, while exhibiting less than 9% in base forgetting. When trained on 100% of labeled base classes with supplementary unlabeled data, SoftER Teacher sets a new standard on semi-supervised few-shot performance using varying amounts of bounding box annotations.

Third, we establish the Label-Efficient Detection benchmark to quantify the utility of unlabeled data for generalized semi-supervised FSOD. We hope that our benchmark serves as a strong baseline, and a blueprint, to inspire future research toward this new problem setting in the community.

## 2 Related Work

**Semi-Supervised Detection.** Recent approaches to SSOD can be summarized into consistency-based and pseudo-labeling categories. The leading consistency method is Humble Teacher [44], which trains a pair of detectors on both labeled and unlabeled data in the student-teacher framework [16, 45]. Humble Teacher learns representations from unlabeled data by enforcing consistency on predicted *soft labels* from region proposals. Humble Teacher was inspired by CSD [18], which was the first approach to leverage consistency regularization [40] for SSOD, but utilizes strong data augmentation to deliver robust performance.

The state of the art on SSOD, however, belongs to a family of pseudo-labeling methods, which trains a pair of detectors on pseudo labels along with (limited) human labels. One such method is Soft Teacher [52] which vastly improves upon its previous counterparts STAC [42] and Unbiased Teacher [29] by enabling end-to-end pseudo-labeling on unlabeled images. More recently, Consistent Teacher [48] advances the performance envelope by reducing the inconsistency of pseudo targets. In both consistency-based and pseudo-labeling methods, the teacher model is an exponential moving average (EMA) of its student counterpart and is used to predict soft or pseudo labels on unlabeled data. The main difference between the two is how the surrogate labels are used to generate unsupervised targets to be jointly trained with the supervised objective.

We extend the strong performance of Soft Teacher by incorporating a new module for Entropy Regression to learn additional representations from unlabeled images by way of region proposals. Our model, aptly named *SoftER Teacher*, combines the attractive benefits of pseudo-labeling with supplementary proposal learning to establish a stronger baseline for SSOD.

**Few-Shot Detection.** Existing methods on FSOD can also be grouped into two categories: meta learning and transfer learning. Early work on meta learning introduced meta models to acquire class-level knowledge for adapting a base detector to novel concepts. Meta learners are jointly trained and fine-tuned with the base detectors to perform tasks like feature re-weighting, such as FSRW [19] and Meta R-CNN [53], or category-specific weight prediction (MetaDet) [49] using few exemplars of support images and ground truth bounding box annotations for the target objects.

Recent work on transfer learning for FSOD found that fine-tuning the last layer of the pre-trained base detector (*i.e.*, the box classifier and regressor) on a balanced subset of base and novel classes, while freezing the rest of the detector, can significantly improve detection accuracy. The simple yet effective two-stage fine-tuning approach TFA [47] outperformed all prior meta learning methods on both base and novel detection metrics. Retentive R-CNN [10] extends TFA by introducing the Bias-Balanced RPN and Re-Detector modules to achieve strong novel performance without sacrificing base accuracy, a desideratum of FSOD. Other transfer learning methods, such as FSCE [43], DeFRCN [36], FADI [3], and DCFS [11], address the shortcomings of the box classifier to boost FSOD. While these methods obtain impressive performance on novel categories, they suffer from considerable base class forgetting, making them sub-optimal in real-world applications requiring efficient and accurate detection on test samples containing instances of both classes.

We show that both base and novel class performances can be further improved when unlabeled images are incorporated into the two-stage fine-tuning procedure without catastrophic base forgetting, which lead to a new standard for semi-supervised FSOD.

**Semi-Supervised Few-Shot Detection.** There have been few attempts at leveraging unlabeled data to improve FSOD, but to our knowledge none directly addressed the task of optimizing for semi-supervised few-shot detection, in which setting both base and novel labels are simultaneously scarce. LVC [21] mines novel targets from the *same base training dataset* via pseudo-labeling to boost novel class detection, but comes at the cost of base performance. UniT [22] obtains impressive results on any-shot detection, but assumes abundant image-level labels for the base and novel targets. And

107 SSFOD [51] performs semi-supervised FSOD within an episodic meta training and $N$-way $k$-shot
108 evaluation framework [20] while also requiring abundant base classes. Our approach is fundamentally
109 different in that we do not strictly depend on abundant base labels, but make effective utilization of
110 *external unlabeled data* for robust semi-supervised FSOD with reduced base degradation.

## 3 Approach

112 We propose to combine the available (limited) labeled examples with supplementary unlabeled images
113 to boost semi-supervised FSOD. We begin with the fully supervised scenario in which we have access
114 to a set of labeled image-target pairs $(x_l, y_l) \in \mathcal{D}_{\text{sup}}$. The supervised FSOD setting [10, 19, 47]
115 assumes a base dataset $C_{\text{base}} \in \mathcal{D}_{\text{sup}}$ with abundant instance-level annotations and a novel dataset
116 $C_{\text{novel}} \in \mathcal{D}_{\text{sup}}$ with only a few $k$ (*e.g.*, $k \in \{1, 5, 10\}$) labeled instances, or "shots", per category. The
117 goal of FSOD is to expand the base detector by adapting it to learn new target concepts such that the
118 resulting detector is optimized for accuracy on a test set comprising both classes in $C_{\text{base}} \cup C_{\text{novel}}$.

119 To maintain parity with existing work, we adopt the simple yet effective two-stage transfer learning
120 approach [10, 47] currently leading the FSOD literature, which comprises an initial stage of base
121 class pre-training followed by a second stage of novel category fine-tuning. We consider the modern
122 Faster R-CNN (FRCN) [37] system as our supervised base detector, which consists of a ResNet [15]
123 *backbone* and a feature pyramid network (FPN) [27] *neck* for feature extraction, a region proposal
124 network (RPN), an RoIAlign [14] operation for mapping proposals to region-of-interest (RoI) features,
125 and a fully-connected *head* for RoI classification and regression. Let FRCN be represented by $f_\theta$, a
126 stochastic function parametrized by a set of learnable weights $\theta$. Formally, the base pre-training step
127 is subjected to the standard supervised objective, over a mini-batch of labeled examples $b_l$, given by:

$$\mathcal{L}_{\text{sup}} = \frac{1}{|b_l|} \sum_{i \in b_l} \mathcal{L}_{\text{cls}}^{\text{roi}}(f_\theta(x_i), y_i) + \mathcal{L}_{\text{reg}}^{\text{roi}}(f_\theta(x_i), y_i). \tag{1}$$

128 Here, $f_\theta(x_i)$ denotes a forward pass on the $i$-th image to produce box classification and localization
129 predictions from class-agnostic proposals, $y_i$ is the $i$-th ground truth annotation containing box labels
130 and coordinates, and $\left(\mathcal{L}_{\text{cls}}^{\text{roi}}, \mathcal{L}_{\text{reg}}^{\text{roi}}\right)$ are the cross-entropy and $L_1$ losses, respectively, for the RoI head.
131 Henceforth for simplicity, we develop our approach only on the RoI head and omit the presentation
132 on the classification and regression losses of the RPN, which remain constant without changes, to
133 predict and localize the "objectness" of region proposals.

### 3.1 What Makes for Effective FSOD?

135 We revisit this question from the perspective of maximizing representation learning while minimizing
136 base forgetting. In two-stage detectors [14, 37], the quality of region proposals is a strong predictor
137 of supervised detection performance [17, 46], since they focus the detector head on candidate RoIs.
138 This is especially true for FSOD approaches based on transfer learning, in which the established
139 procedure is to freeze the RPN during few-shot fine-tuning. Intuitively, if we can incorporate methods
140 and/or data to boost representation learning by way of the RPN, then the detector should have a
141 higher chance of discovering novel categories to improve few-shot performance.

142 We conduct extensive experiments on the COCO dataset to verify our intuition. We split the dataset
143 into disjoint 60 base and 20 novel categories and pre-train three variants of the FRCN detector on
144 the base classes: (i) FRCN-Base, (ii) FRCN-Base augmented with COCO `unlabeled2017` images
145 leveraging the Soft Teacher formulation, and (iii) FRCN-Full using both base and novel classes to
146 represent the upper-bound performance. We also experiment with CRPN-Base, a method specially
147 designed to improve proposal quality and detection performance using a two-stage Cascade RPN [46].

148 Figure 3a quantifies the "class-agnosticism" of various RPNs, using the standard metric `AR@300`
149 proposals, for varying fractions of base labels. Surprisingly, unlabeled data has the remarkable ability
150 to boost proposal recall on novel-only categories, even in the extremely low 1% label limit. Somewhat
151 unsurprising is the ability of `CRPN-Base` to propose novel objects competitive with `FRCN-Base`
152 + `Unlabeled` when more base labels are available. Consistent with previous findings [10, 21],
153 Figures 3b and 3c show that the vanilla supervised `FRCN-Base` has a strong tendency to reject novel
154 objects as background, due to the lack of annotations, resulting in the worst recall on novel classes.

155 As alluded in Section 1, the contribution of unlabeled data to FSOD goes beyond improving base
156 and novel detection performances; unlabeled data can also help mitigate catastrophic base forgetting.

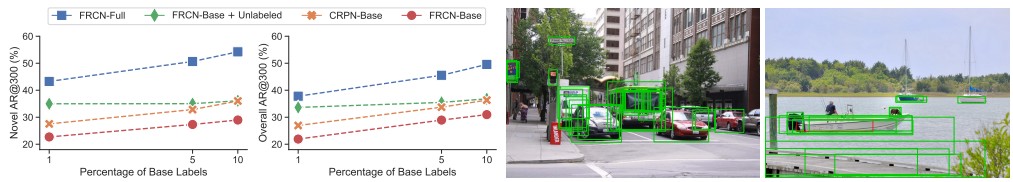

**(a)** Impact of unlabeled data on proposal quality    **(b)** RPN w/ 1% labels    **(c)** RPN w/ 10% labels

**Figure 3:** We analyze the effectiveness of the RPN as a function of base labels. **(a)** Unlabeled data provides a convincing boost in proposal quality, closing the gap between the `Base` and `Full` detectors, which should lead to better discovery of novel categories during fine-tuning. **(b–c)** In low-label regimes, unlabeled data can help produce diverse proposals (green boxes) on novel unseen objects {`boat`, `bus`, `car`, `dog`}, whereas the vanilla supervised `FRCN-Base` fails to capture comparable foreground objects (red boxes). Best viewed digitally.

We find analogous effectiveness of `FRCN-Base + Unlabeled` on the combined `Overall AR@300` metric, for both base + novel objects, suggesting the RPN, when trained with unlabeled data, has the ability to retain base proposals and help combat base degradation during few-shot fine-tuning.

**Discussion.** This paper rethinks a different and more versatile way to improve the RPN for FSOD while avoiding catastrophic forgetting. The previous LVC approach proposed to unfreeze the RPN during fine-tuning to obtain large performance gains on novel categories, but comes at the cost of significant base degradation (up to 19%). Similarly, FSCE [43] proposed to unfreeze the RPN while also doubling the number of proposals to encourage novel foreground detection during fine-tuning. However, this method increases the detection overhead and remains unclear whether it helps mitigate base forgetting. We illustrate that simply adding unlabeled data to the base detector leads to a compelling boost in proposal quality, without the need for any *ad hoc* modifications to the RPN.

We attribute this unique benefit of our approach to the potential base-novel object interactions found in abundant images. When learning with unlabeled data, the base detector can obtain semantically similar cues of novel objects to inform the RPN on foreground detection. Sun *et al.* [43] showed that visually analogous objects have high cosine similarity scores (*e.g.*, sim(`cow`, `horse`) = 0.39). With 1% of labels, these base-novel interactions are limited, resulting in a recall of 22.7%. Given a sizable unlabeled dataset, the base detector improves its representations to yield a gain of +12.3 points.

## 3.2 Semi-Supervised Base Pre-Training

Motivated by the promising utility of unlabeled data, we now relax the strict assumption on having abundant base classes for FSOD and introduce a new and more general setting of having a small fraction of base labels given abundant unlabeled images. We revisit the task of semi-supervised base pre-training by formulating an unsupervised loss computed on an unlabeled dataset $\mathcal{D}_{\text{unsup}}$ to be jointly trained with the supervised loss on $\mathcal{D}_{\text{sup}}$. We consider the following canonical optimization objective widely adopted as part of the framework for semi-supervised learning [1, 23, 33]:

$$\min_{\theta} \mathcal{L}_{\text{sup}}(\mathcal{D}_{\text{sup}}, \theta) + \lambda \mathcal{L}_{\text{unsup}}(\mathcal{D}_{\text{unsup}}, \theta), \tag{2}$$

where $\lambda > 0$ is a hyper-parameter controlling the contribution of the unsupervised component. Next, we describe the unsupervised criterion on $\mathcal{D}_{\text{unsup}}$ to make FRCN into a semi-supervised detector.

**Soft Teacher.** We adopt Soft Teacher [52] as the baseline SSOD formulation for its simplicity but strong performance. Soft Teacher trains FRCN in a student-teacher fashion on both labeled and unlabeled data. The student is trained on labeled examples in the standard supervised manner per Eq. (1). For unlabeled images, the teacher is treated as a fixed detector to generate thousands of box candidates, most of which are eliminated for redundancy with non-maximum suppression. Additionally, box candidates are thresholded for foreground objects and go through an iterative jittering-refinement procedure to reduce localization variance, resulting in a set of high-quality pseudo boxes to be jointly trained with ground truth annotations.

As is common practice [41, 45], the teacher's parameters $\bar{\theta}$ are updated from the student's via $\bar{\theta} = \text{EMA}(\theta)$ at each training step. Integral to the success of Soft Teacher is a student-teacher data augmentation strategy inspired by STAC [42]. The student trains on unlabeled images subjected to complex random perturbations, akin to RandAugment [6], including affine transforms. Separately,

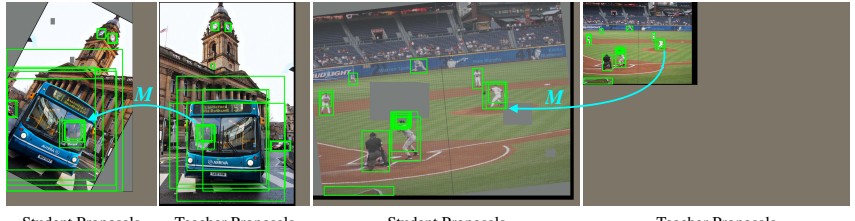

| Student Proposals | Teacher Proposals | Student Proposals | Teacher Proposals |

**Figure 4:** Visualization of student-teacher proposals with confidence scores $\geq 0.99$. As illustrated by the arrow, a pair of student-teacher proposals is related by a transformation matrix $M$, which is used to align proposals between student and teacher images for enforcing box classification similarity and localization consistency.

the teacher receives weakly augmented images with simple random resizing and flipping. This multi-stream augmentation design allows the teacher to generate reliable unsupervised targets on easy images to guide the student's learning on difficult images for better generalization.

At the time of box classification and regression in the RoI head, we have a set of unlabeled images along with teacher-generated pseudo labels $(x_u, \hat{y}_u) \in \mathcal{D}_{\text{unsup}}$. The unsupervised loss for Soft Teacher on a mini-batch of unlabeled images $b_u$ is defined as:

$$\mathcal{L}_{\text{unsup}}^{\text{soft}} = \frac{1}{|b_u|} \sum_{i \in b_u} \mathcal{L}_{\text{cls}}^{\text{roi}}(f_\theta(x_i), \hat{y}_i) + \mathcal{L}_{\text{reg}}^{\text{roi}}(f_\theta(x_i), \hat{y}_i). \tag{3}$$

**SoftER Teacher.** The design of Soft Teacher employs class confidence thresholding and box jittering to select high-quality pseudo-label candidates for unsupervised classification and regression. However, it uses an aggressive threshold of 0.9, resulting in a trade-off between low recall and high precision at 33% and 89%, respectively [52]. We observe that low recall can result in poor detection performance on small and ambiguous objects [26], especially in low-label regimes where the teacher has insufficient confidence about its predicted pseudo labels. We aim to extend Soft Teacher and improve its detection recall by learning additional representations from abundant region proposals.

Given a set of proposals $p$ generated by the student's RPN on a batch of unlabeled images, we apply the student-teacher data augmentation pipeline described above to obtain $(p_s, p_t)$, denoting transformed student and teacher proposals, which are related to each other by a transformation matrix $M$. We then forward pass Faster R-CNN twice, as the student $f_\theta$ and teacher $f_{\bar{\theta}}$, to obtain two sets of RoI outputs for predicted box classification logits $(z_s, z_t)$ and localization coordinates $(r_s, r_t)$. Let $g_c$ be the softmax function over the channel dimension $c$. We define an auxiliary unsupervised criterion for proposal box similarity based on a cross-entropy measure $H(z_s, z_t)$:

$$\mathcal{L}_{\text{cls}}^{\text{ent}} = \frac{1}{\sum_i w_i} \sum_{i \in p} w_i \cdot H(z_{is}, z_{it}),$$
$$\text{in which} \quad H(z_{is}, z_{it}) = -\frac{1}{C} \sum_{c \in C} g_c(z_{it}) \log g_c(z_{is}). \tag{4}$$

Here, $g_c$ outputs a distribution over $C$ classes and $w_i$ is the Boolean weight for the predicted positive (foreground) class: $w_i = 1$ if $\text{argmax}(z_{it}) \neq background$, else $w_i = 0$.

Similarly for proposal box regression, we constrain the predicted box coordinates $(r_s, r_t)$ to be close. Since there are complex geometric distortions between the two, we first map teacher proposal coordinates $r_t$ to the student space using the transformation $M$. Then, we align the proposal boxes via the intersection-over-union (IoU) criterion:

$$\mathcal{L}_{\text{reg}}^{\text{iou}} = 1 - \frac{1}{|p|} \sum_{i \in p} w_i \cdot \text{IoU}(r_{is}, M(r_{it})), \tag{5}$$

where we treat the IoU metric as a loss [38] to quantify the discrepancy between student and teacher proposal coordinates. Note that both the cross-entropy and IoU losses, Eqs. (4) and (5), are computed only on predicted foreground classes.

Recall that we have two different transformation pipelines operating on each proposal, so we have two augmented views of each proposal. Figure 4 illustrates that by enforcing these randomly

augmented views, and their box coordinates, to be *similar*, we enable the student to tap into abundant region proposals to learn diverse feature representations across a spectrum of scale, color, and geometric perturbations. Our formulation draws inspiration from recent research on self-supervised representation learning with multi-augmented views [5, 13]. Note the cross-entropy similarity between the student and teacher predictions, Eq. (4), can be interpreted as a form of entropy regularization [12], which has been proven to work well in various semi-supervised classification scenarios [33, 34]. The overall optimization objective at the RoI head for our SoftER Teacher model is computed as:

$$\mathcal{L}_{\text{total}}^{\text{softer}} = \mathcal{L}_{\text{sup}} + \alpha \mathcal{L}_{\text{unsup}}^{\text{soft}} + \beta \left( \mathcal{L}_{\text{cls}}^{\text{ent}} + \mathcal{L}_{\text{reg}}^{\text{iou}} \right), \tag{6}$$

where we set $\alpha = \frac{|b_u|}{|b_l|}$ following Soft Teacher and find $\beta = 2\alpha$ works well across all experiments.

**Discussion.** Our approach to proposal learning is unique and different from Humble Teacher [44] in two major ways: (1) we use diverse data transformations, including geometric distortions, to enforce proposal similarity, whereas Humble Teacher considered only simple random flipping and color transforms; and (2) we adopt the IoU metric for box localization consistency over $L_2$ distance, which has been shown to produce inferior box regression [38]. Further, Humble Teacher acknowledged that matching student-teacher proposals under complex affine transforms is not a trivial task and "could lead to undesirably complicated details." We solve this complicated task by tracking the affine matrix $M$ in our entropy regression module, to address a key weakness of Soft Teacher by boosting object recall, thereby enabling SoftER Teacher to demonstrate superior learning with unlabeled data over Humble Teacher and Soft Teacher. Please refer to Appendix B.3 for detailed comparative results.

### 3.3 Semi-Supervised Few-Shot Fine-Tuning

We propose a simple two-step approach to harness unlabeled data for semi-supervised few-shot fine-tuning. First, we initialize the few-shot detector, $f'_{\theta} \leftarrow f_{\bar{\theta}}$, with parameters copied from the base *teacher* detector pre-trained with unlabeled data per Eq. (6). And second, we further train the RoI head of $f'_{\bar{\theta}}$ on novel classes using the available few-shot and unlabeled examples while freezing the base backbone, FPN, and RPN components. Then, we fine-tune the few-shot detector on a balanced training set of $k$ shots per class containing both base and novel instances. We only update the RoI box classifier while freezing all other components, including the box regressor, since it is the main source of error [11, 43]. To our knowledge, *we are the first to incorporate external unlabeled data with few-shot fine-tuning*, which provides a compelling boost to novel performance while enjoying substantial gains in base detection without catastrophic forgetting. We present detailed ablation studies in Appendix A to validate our approach and design choices.

## 4 Experiments

**Datasets.** Consistent with the current literature on FSOD, we evaluate our approach on the challenging PASCAL VOC [9] and MS-COCO 2017 [28] detection benchmarks. For VOC, we use the combined VOC07+12 `trainval` splits as the labeled training set and evaluate performance on the VOC2007 `test` set. For COCO, we utilize the `train2017` split as labeled data and test on `val2017`. We also leverage `COCO-20`, the subset of COCO data having the same 20 class names as VOC, and COCO `unlabeled2017` as the sources of supplementary unlabeled data.

**Performance Metrics.** Following established evaluation protocol, we assess detection performance using $\text{AP}_{50}$ for the average precision at overlap threshold 0.5 and $\text{AP}_{50:95}$ for the mean average precision computed over a range of 10 overlap thresholds between 0.5 and 0.95. We also report *average recall AR* to complement AP for assessing object coverage, which has previously been used to evaluate detection performance of small objects [24, 26].

**Implementation Details.** For most experiments, we adopt the ResNet-101 backbone pre-trained on ImageNet 1K [8] for a direct comparison with existing work. For some experiments, we also employ ResNet-50 to demonstrate parameter-efficient learning with SoftER Teacher. We implement our models in MMDetection [4] and PyTorch [35]. Complete details are given in Appendix C.

### 4.1 SoftER Teacher is a Parameter- and Label-Efficient Few-Shot Detector

We conduct our few-shot experiments on the same VOC and COCO samples provided by the TFA benchmark [47]. The VOC dataset is randomly partitioned into 15 base and 5 novel classes, in which

**Table 1:** FSOD results evaluated on COCO `val2017`. We report the mean and 95% confidence interval over 5 random samples for our models. SoftER Teacher with ResNet-101 is the best model on the combined `Overall` AP metric, incurring less than 9% in base forgetting *vs.* 11%–DCFS, 17%–DeFRCN, and 19%–LVC.

| COCO `val2017` Method | Backbone | Base AP$_{50:95}$ | Base AP$_{50:95}$ (60 Classes) | | | Novel AP$_{50:95}$ (20 Classes) | | | Overall AP$_{50:95}$ (80 Classes) | | |
|---|---|---|---|---|---|---|---|---|---|---|---|
| | | | 5-Shot | 10-Shot | 30-Shot | 5-Shot | 10-Shot | 30-Shot | 5-Shot | 10-Shot | 30-Shot |
| LVC [21] | R-50 | – | – | 29.7 ± n/a | 33.3 ± n/a | – | 17.6 ± n/a | 25.5 ± n/a | – | 26.7 ± n/a | 31.4 ± n/a |
| SoftER Teacher (Ours) | R-50 | 42.0 | 38.4 ± 0.2 | 38.4 ± 0.2 | 39.7 ± 0.2 | 8.2 ± 0.3 | 10.3 ± 0.5 | 12.9 ± 0.6 | 30.9 ± 0.1 | 31.4 ± 0.2 | 33.0 ± 0.1 |
| TFA [47] | R-101 | 39.3 | 32.3 ± 0.6 | 32.4 ± 0.6 | 34.2 ± 0.4 | 7.0 ± 0.7 | 9.1 ± 0.5 | 12.1 ± 0.4 | 25.9 ± 0.6 | 26.6 ± 0.5 | 28.7 ± 0.4 |
| LVC [21] | R-101 | 39.3 | – | 31.9 ± n/a | 33.0 ± n/a | – | 17.8 ± n/a | 24.5 ± n/a | – | 28.4 ± n/a | 30.9 ± n/a |
| DeFRCN [36] | R-101 | 39.3 | 32.6 ± 0.3 | 34.0 ± 0.2 | 34.8 ± 0.1 | 13.6 ± 0.7 | 16.8 ± 0.6 | 21.2 ± 0.4 | 27.8 ± 0.3 | 29.7 ± 0.2 | 31.4 ± 0.1 |
| DCFS [11] | R-101 | 39.3 | 35.0 ± 0.2 | 35.7 ± 0.2 | 35.8 ± 0.2 | 15.7 ± 0.5 | 18.3 ± 0.4 | 21.9 ± 0.3 | 30.2 ± 0.2 | 31.4 ± 0.2 | 32.3 ± 0.2 |
| Retentive R-CNN [10] | R-101 | 39.3 | 39.3 ± n/a | 39.2 ± n/a | 39.3 ± n/a | 7.7 ± n/a | 9.5 ± n/a | 12.4 ± n/a | 31.4 ± n/a | 31.8 ± n/a | 32.6 ± n/a |
| SoftER Teacher (Ours) | R-101 | 44.4 | 40.3 ± 0.2 | 40.2 ± 0.3 | 41.4 ± 0.2 | 8.7 ± 0.6 | 11.0 ± 0.4 | 14.0 ± 0.6 | 32.4 ± 0.2 | 32.9 ± 0.1 | 34.6 ± 0.1 |

**Table 2:** FSOD results evaluated on VOC07 `test`. We report the mean and 95% confidence interval over 10 random samples for our models. SoftER Teacher with ResNet-50 surpasses the supervised MPSR, TFA, and Retentive R-CNN models with ResNet-101 by a large margin on most metrics under consideration, while being more parameter-efficient. Results for the other two partition splits are given in Appendix B.1.

| VOC07 `test` – Split 1 Method | Backbone | Base AP$_{50}$ | Base AR$_{50}$ | Base AP$_{50}$ (15 Classes) | | | Novel AP$_{50}$ (5 Classes) | | | Overall AP$_{50}$ (20 Classes) | | |
|---|---|---|---|---|---|---|---|---|---|---|---|---|
| | | | | 1-Shot | 5-Shot | 10-Shot | 1-Shot | 5-Shot | 10-Shot | 1-Shot | 5-Shot | 10-Shot |
| MPSR [50] | R-101 | 80.8 | – | 61.5 | 69.7 | 71.6 | 42.8 | 55.3 | 61.2 | 56.8 | 66.1 | 69.0 |
| Retentive R-CNN [10] | R-101 | 80.8 | – | 80.9 | 80.8 | 80.8 | 42.4 | 53.7 | 56.1 | 71.3 | 74.0 | 74.6 |
| TFA [47] | R-101 | 80.8 | – | 77.6 ± 0.2 | 77.4 ± 0.3 | 77.5 ± 0.2 | 25.3 ± 2.2 | 47.9 ± 1.2 | 52.8 ± 1.0 | 64.5 ± 0.6 | 70.1 ± 0.4 | 71.3 ± 0.3 |
| Faster R-CNN (Our Impl.) | R-50 | 81.7 | 88.0 | 82.0 ± 0.2 | 82.4 ± 0.1 | 82.3 ± 0.1 | 27.9 ± 3.2 | 52.1 ± 2.1 | 58.2 ± 1.6 | 68.5 ± 0.8 | 74.9 ± 0.5 | 76.2 ± 0.4 |
| Soft Teacher (Our Impl.) | R-50 | 85.3 | 91.2 | 84.5 ± 0.3 | 85.2 ± 0.1 | 85.2 ± 0.1 | 29.5 ± 4.2 | 56.2 ± 2.6 | 62.3 ± 1.8 | 70.8 ± 1.1 | 78.0 ± 0.7 | 79.5 ± 0.5 |
| SoftER Teacher (Ours) | R-50 | 85.9 | 92.5 | 84.5 ± 0.4 | 85.5 ± 0.1 | 85.5 ± 0.1 | 31.6 ± 3.9 | 57.7 ± 2.6 | 63.4 ± 1.7 | 71.3 ± 1.2 | 78.5 ± 0.7 | 80.0 ± 0.4 |

there are $k \in \{1, 5, 10\}$ labeled boxes per category sampled from the joint VOC07+12 `trainval` splits. And the COCO dataset is divided into 60 base and 20 novel classes having the same VOC category names with $k \in \{5, 10, 30\}$ shots. We leverage `COCO-20` and `unlabeled2017` as external unlabeled data to augment base pre-training and novel fine-tuning on VOC and COCO, respectively.

Table 1 compares the effectiveness of SoftER Teacher against transfer-learning methods representing the state of the art on COCO, for the evaluation of both base and novel performances. We report the ideal supervised base AP of 39.3 [10, 47] along with our substantially improved semi-supervised base AP of 44.4 to measure the extent of base forgetting. Recall that the more realistic evaluation metric for *generalized* FSOD is not only novel AP, but the combination of base and novel AP. We summarize the following key takeaways: **(a)** SoftER Teacher with ResNet-101 trained with supplementary unlabeled data is the best model on the combined `Overall` AP metric for 80 classes, leading the next best Retentive R-CNN by up to +2.0 AP; **(b)** SoftER Teacher with ResNet-50 is on par with Retentive R-CNN while being more parameter-efficient; and **(c)** *SoftER Teacher achieves the state of the art while being more efficient with respect to parameters and labels*.

We notice a recurring theme in the FSOD literature that seems to favor novel class performance while ignoring base accuracy, even though the detection of both classes is equally important at test time. Table 2 shows comparative results on VOC, for which there are few existing work evaluating on both base and novel AP. We report the ideal supervisd base AP of 80.8 [10, 47] along with our vastly expanded base AP of 85.9 using unlabeled data. SoftER Teacher with ResNet-50 incurs negligible base forgetting of less than 1.6% while exceeding MPSR, TFA, and Retentive R-CNN with ResNet-101 by a notable margin on most metrics. Although MPSR achieves impressive one-shot performance on novel categories, it suffers catastrophic base forgetting by as much as 24%. Retentive R-CNN does not exhibit base class degradation, but generally falls behind on novel class performance. Tables 1 and 2 corroborate our observation on the trade-off between novel performance and base forgetting, for which our approach aims to simultaneously optimize. Due to page limit, **we refer to Appendix A for detailed ablation studies analyzing the design and benefits of SoftER Teacher.**

### 4.2 How Does Proposal Quality Impact Semi-Supervised Few-Shot Detection?

In this section, we continue our discussion from Sec. 3.1 by analyzing semi-supervised FSOD as a function of proposal quality in Figure 5. We measure proposal quality using the metric AR@$p$ [46], for $p \in \{100, 300, 1000\}$ proposals, averaged over thresholds between 0.5 and 0.95. We arrive at the following conclusions: **(a)** SoftER Teacher produces better proposals than the comparisons across varying fractions of labels; and **(b–d)** proposal quality is strongly correlated with semi-supervised FSOD, an insightful empirical finding that extends existing results beyond supervised detection [17].

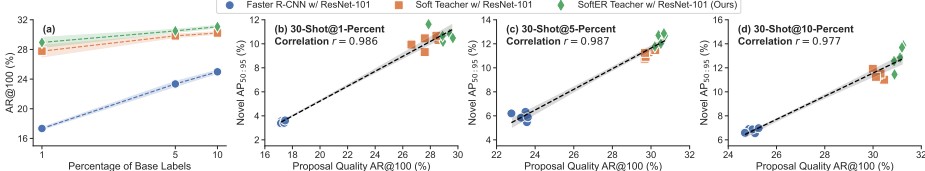

**Figure 5:** Proposal quality is highly correlated with semi-supervised FSOD. SoftER Teacher produces the best proposals among the comparisons **(a)**, which yield the strongest 30-shot performances **(b–d)**. Shaded regions denote standard deviation over 5 samples. Appendix B.2 gives similar trends for 5-shot and 10-shot results.

**Table 3:** We introduce the Label-Efficient Detection benchmark for generalized semi-supervised FSOD on COCO. All models are trained with ResNet-101. We report the mean and standard deviation over 5 samples. Using only 10% of base labels **(bottom row)**, SoftER Teacher surpasses the supervised novel performance of Retentive R-CNN trained with 100% of base labels **(top row)** while incurring less than 9% in base degradation.

| Method | % Labeled | Base AP$_{50:95}$ | Base AR$_{50:95}$ | Base AP$_{50:95}$ (60 Classes) | | | Novel AP$_{50:95}$ (20 Classes) | | | Overall AP$_{50:95}$ (80 Classes) | | |
|---|---|---|---|---|---|---|---|---|---|---|---|---|
| | | | | 5-Shot | 10-Shot | 30-Shot | 5-Shot | 10-Shot | 30-Shot | 5-Shot | 10-Shot | 30-Shot |
| Retentive R-CNN | 100 | 39.3 | – | 39.3 | 39.2 | 39.3 | 7.7 | 9.5 | 12.4 | 31.4 | 31.8 | 32.6 |
| SoftER Teacher | | 44.4 | 56.1 | $40.3 \pm 0.2$ | $40.2 \pm 0.3$ | $41.4 \pm 0.2$ | $8.7 \pm 0.6$ | $11.0 \pm 0.4$ | $14.0 \pm 0.6$ | $32.4 \pm 0.2$ | $32.9 \pm 0.1$ | $34.6 \pm 0.1$ |
| Faster R-CNN | 1 | $8.7 \pm 0.3$ | $12.3 \pm 0.5$ | $9.8 \pm 0.3$ | $10.0 \pm 0.4$ | $10.8 \pm 0.3$ | $1.9 \pm 0.3$ | $2.7 \pm 0.1$ | $3.5 \pm 0.1$ | $7.8 \pm 0.2$ | $8.2 \pm 0.3$ | $9.0 \pm 0.2$ |
| Soft Teacher | | $19.9 \pm 1.0$ | $30.7 \pm 1.1$ | $19.4 \pm 0.7$ | $19.9 \pm 0.8$ | $21.2 \pm 0.7$ | $5.9 \pm 0.8$ | $7.9 \pm 0.7$ | $10.1 \pm 0.5$ | $16.0 \pm 0.6$ | $16.9 \pm 0.7$ | $18.4 \pm 0.6$ |
| SoftER Teacher | | $19.8 \pm 0.9$ | $32.5 \pm 1.0$ | $19.2 \pm 0.6$ | $19.8 \pm 0.5$ | $21.1 \pm 0.5$ | $6.7 \pm 0.3$ | $8.8 \pm 0.2$ | $10.8 \pm 0.5$ | $16.1 \pm 0.5$ | $17.1 \pm 0.4$ | $18.5 \pm 0.5$ |
| Faster R-CNN | 5 | $19.1 \pm 0.3$ | $25.6 \pm 0.4$ | $18.5 \pm 0.5$ | $18.9 \pm 0.3$ | $20.0 \pm 0.5$ | $3.5 \pm 0.2$ | $4.6 \pm 0.2$ | $5.9 \pm 0.3$ | $14.8 \pm 0.4$ | $15.3 \pm 0.2$ | $16.5 \pm 0.4$ |
| Soft Teacher | | $29.6 \pm 0.3$ | $38.7 \pm 0.3$ | $27.5 \pm 0.4$ | $27.8 \pm 0.5$ | $29.2 \pm 0.5$ | $6.7 \pm 0.7$ | $8.9 \pm 0.4$ | $11.1 \pm 0.3$ | $22.3 \pm 0.4$ | $23.1 \pm 0.3$ | $24.7 \pm 0.4$ |
| SoftER Teacher | | $30.2 \pm 0.2$ | $40.7 \pm 0.3$ | $27.5 \pm 0.4$ | $27.9 \pm 0.4$ | $29.3 \pm 0.2$ | $7.9 \pm 0.4$ | $10.1 \pm 0.5$ | $12.4 \pm 0.5$ | $22.6 \pm 0.3$ | $23.4 \pm 0.3$ | $25.1 \pm 0.2$ |
| Faster R-CNN | 10 | $24.7 \pm 0.2$ | $32.8 \pm 0.3$ | $22.6 \pm 0.4$ | $22.8 \pm 0.1$ | $24.2 \pm 0.2$ | $3.8 \pm 0.5$ | $5.3 \pm 0.2$ | $6.8 \pm 0.2$ | $17.9 \pm 0.3$ | $18.4 \pm 0.1$ | $19.9 \pm 0.2$ |
| Soft Teacher | | $33.3 \pm 0.2$ | $42.4 \pm 0.2$ | $30.5 \pm 0.5$ | $30.7 \pm 0.4$ | $32.1 \pm 0.3$ | $6.8 \pm 0.3$ | $9.0 \pm 0.6$ | $11.4 \pm 0.3$ | $24.6 \pm 0.4$ | $25.3 \pm 0.4$ | $26.9 \pm 0.3$ |
| SoftER Teacher | | $33.4 \pm 0.4$ | $44.1 \pm 0.2$ | $30.3 \pm 0.5$ | $30.6 \pm 0.5$ | $32.0 \pm 0.4$ | $7.9 \pm 1.3$ | $10.4 \pm 1.1$ | $12.9 \pm 1.0$ | $24.6 \pm 0.1$ | $25.6 \pm 0.3$ | $27.2 \pm 0.3$ |

Although the strong Soft Teacher baseline is effective at harnessing unlabeled data for semi-supervised FSOD, *our approach demonstrates superior learning by addressing a key shortcoming of Soft Teacher*. SoftER Teacher boosts object recall via our proposed Entropy Regression module, improving on Soft Teacher by $+1.2$ base AR@100, which yields a gain of $+1.5$ novel AP for the 30-Shot@10-Percent setting. These results further strengthen our empirical observation that a stronger semi-supervised detector leads to a more label-efficient few-shot detector. Future work would explore if our finding can be extended to a more general case with other SSOD formulations including one-stage detectors.

### 4.3 A New Benchmark for Generalized Semi-Supervised Few-Shot Detection

We present our Label-Efficient Detection benchmark on COCO `val2017` in Table 3, evaluated using the supervised Faster R-CNN baseline along with the semi-supervised Soft Teacher and SoftER Teacher models. Our protocol for semi-supervised FSOD is as follows. In the first stage, we pre-train the base detector on the disjoint 60 base categories using $\{1, 5, 10\}$ percent of labels per Eq. (6). In the second stage, we transfer the parameters of the base teacher detector to the few-shot detector and fine-tune its RoI box classifier, keeping other components frozen, on a balanced training set of $k \in \{5, 10, 30\}$ shots per class containing both base and novel examples. In both stages, we supplement base pre-training and novel fine-tuning with images from COCO `unlabeled2017`.

We report AP$_{50:95}$ performances on both base and novel classes along with the aggregated overall metric. We also report the ideal AP$_{50:95}$ and AR$_{50:95}$ metrics obtained from the first stage of base pre-training to measure the potential for base forgetting during the few-shot fine-tuning step. We encourage future work to follow suit as we emphasize the importance of optimizing for accuracy on both base and novel classes, a desideratum of generalized few-shot object detection.

## 5 Conclusion

This paper presented the Label-Efficient Detection framework to quantify the utility of unlabeled data for generalized semi-supervised FSOD. Central to the framework is our SoftER Teacher, a robust detector combining the strengths of pseudo-labeling with representation learning on unlabeled images. We demonstrated two main areas of impact: (1) SoftER Teacher achieves superior learning with unlabeled data to boost semi-supervised FSOD without relying on an abundance of labels; and (2) our framework sheds empirical insight into a potential relationship that a stronger semi-supervised detector leads to a more effective few-shot detector, the basis of which could inspire future research.

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
