# OpenReview forum: "Boosting Semi-Supervised Few-Shot Object Detection with SoftER Teacher"
_NeurIPS.cc/2023/Conference — Submitted to NeurIPS 2023_

### Official Review · Reviewer_F1pt · 2023-06-30

**Soundness:** 3 good
**Presentation:** 2 fair
**Contribution:** 1 poor
**Rating:** 4
**Confidence:** 5

**Summary:**

This paper studies a new task named semi-supervised few-shot object detection, where both of base and novel classes are supposed to be scarce.
The author first finds the vanilla supervised FRCN trained on base classes has a low recall on novel classes, and trained with extra unlabeled novel data can effectively improve the novel recall.
Then the author follows the SSOD framework Soft Teacher to do semi-supervised base training, where only partial base data is available. However, the original Soft Teacher has a low recall of small and ambiguous objects. Thus the author proposes a new proposal learning method to improve it.
Finally, the pre-trained model is then semi-supervised fine-tuned on a balance set comprised of both base and novel samples.
Experiments show Softer Teacher achieves a good GFSOD performance.

**Strengths:**

* The idea is straightforward and has good soundness.
* The method has promising results in the GFSOD setting.

**Weaknesses:**

1. The proposed task of semi-supervised few-shot objects is similar to semi-supervised object detection. Since both base and novel classes are scarce, what's the meaning of splitting classes into base and novel? I can't see any practical significance in this task.

2. From my point of view, the setting proposed in the paper is closer to a semi-supervised than a few-shot object detection problem. Particularly, one of the key properties of few-shot learning is that the model does not know the novel classes in training, so it can adapt quickly to new classes with only a few examples in testing either using meta-learning (e.g., meta-RCNN) or small-#step finetuning (e.g., TFA).
Another common sense in FSOD is that the novel classes are authentically rare, and we cannot find more images about that class, regardless of label or unlabeled.
Therefore, the proposed approach works best for semi-supervised object detection rather than few-shot object detection, and it is not fair to compare it with FSOD works.

3. The proposed method SoftER Teacher is an incremental improvement based on existing work SSOD Soft Teacher. The only improvement seems to be that the author appends a new loss to constrain outputs from the teacher and the student should be close to corresponding proposals, but the method is more likely to be only related to semi-supervised learning, it seems nothing about few-shot learning.

4. Section 3.1 is named "What makes for Effective FSOD", but 3.1 studies unlabeled data can improve the novel recall of the FSOD model. The title is not very accurate.

5. In line 252, the author argues, "we are the first to incorporate external unlabeled data with few-shot fine-tuning", I don't think it is a contribution or anything good.

6. There is no component analysis or ablation experiments to demonstrate the effectiveness of the proposed method. For example, the performance comparison of w/wo  the proposal learning loss.

7. The performance improvement is minor in the FSOD setting (not GFSOD). The novel performance is actually bad. The superior GFSOD performance may attribute to the strong baseline Softer Teacher on base performance.

8. The authors do not show any numbers related to the training resources (memory and time).

**Questions:**

1. What's the difference between the proposed semi-supervised few-shot objection and semi-supervised object detection? What's the practical meaning or practical application scenario that semi-supervised few-shot objection covers but semi-supervised object detection does not?

2. Besides the proposal learning loss, any more improvements upon the baseline framework Soft Teacher? Where is the ablation that demonstrates its effectiveness of it?

3. SoftER teacher adopts FRCN for both teacher and student, how about the performance when replacing the FRCN with other FSOD methods like DeFRCN?

---

> ### Author Rebuttal · Authors · 2023-08-09
>
> ### We thank Reviewer `F1pt` for your constructive feedback. Please find below our responses, along with additional experiments, to address your questions and concerns.
> ### 1. What's the difference between the proposed semi-supervised FSOD and SSOD?
> The current leading approach for FSOD is the two-stage procedure comprising the first stage of representation learning via base pre-training followed by few-shot target adaptation via transfer learning. Our work introduces supplementary unlabeled data in both stages, resulting in a new semi-supervised FSOD setting, to substantially improve base and novel class performances while also mitigating base forgetting. Reviewers `F1pt` and `Qe4L` pose a valid question: can we merge the two stages into one and train under the SSOD setting?
>
> Recall in FSOD benchmarks, the base dataset is assumed to be fully annotated with boxes covering all instances of interest, whereas the novel set is sparsely labeled at $k$-shots, or bounding boxes, per category. So a novel image containing 3 cats may not be fully labeled but only one box (1-shot) is annotated while the other two ignored. Thus, the benefit of the two-stage approach is to separate fully annotated base examples from sparsely annotated novel ones, thereby allowing for algorithms to optimize on both classes.
>
> The reviewer is correct in that the first stage is aligned with SSOD, where we train on {1,5,10} percent of base classes with unlabeled data. However, the second stage is needed to adapt the base domain to novel concepts while preserving base performance by freezing the appropriate layers. We conduct an experiment in the table below to show that if we approach the FSOD problem as SSOD, denoted as "One-Stage Semi-Supervised", we would run into the issue of foreground objects being rejected as background (i.e., missing labels), resulting in a drastic reduction in base performance.
>
> |Training Protocol|10-Shot bAP|30-Shot bAP|10-Shot nAP|30-Shot nAP|
> |-|-|-|-|-|
> |One-Stage Semi-Supervised|11.0|16.9|11.6|15.3|
> |Proposed Two-Stage Few-Shot (Section 3)|37.2|38.6|10.6|12.3|
>
> ### 2. The proposed SoftER Teacher is an incremental improvement on existing Soft Teacher for SSOD. It seems nothing about few-shot learning.
> L234-243 in Section 3.2 argue that SoftER Teacher contributes a non-trivial extension to Soft Teacher with our Entropy Regression (ER) module for proposal learning with complex affine transforms, which has not been attempted before. ER addresses a key weakness of Soft Teacher by enhancing proposal recall, which translates to convincingly better FSOD (Figure 5). ***For semi-supervised FSOD (Table 3), SoftER Teacher improves on Soft Teacher by +1.7 base class AR, which yields a gain of up to +1.5 novel class AP. These results support our empirical finding in Section 4.2 demonstrating a potential relationship between SSOD and FSOD.***
>
> ### 3. Section 3.1 is named "What Makes for Effective FSOD", but 3.1 studies unlabeled data can improve the novel recall of the FSOD model. The title is not very accurate.
> Section 3.1 presents a new empirical analysis linking the effectiveness of FSOD to unlabeled data by way of proposal recall. Then, we follow up with extensive experiments in Section 4.2 to bolster our analysis and claims. Thus, we believe Section 3.1 is aptly titled within the context of our premise and conclusion.
>
> ### 4. There is no ablation experiments to demonstrate the effectiveness of the proposed method.
> Due to page limit, we put three detailed ablation studies analyzing the design and benefits of SoftER Teacher in Appendix A of the supplementary material, and refer to them throughout the main paper (L254 and L299). For your convenience, we reproduce Table 4 from Appendix A.1 below. We will include the ablation studies in the camera-ready version using the extra page.
>
> |Proposal Similarity Measure|Proposal IoU Regression?|AP|AR|
> |-|-|-|-|
> |None|No|22.4|30.8|
> |KL-Divergence|No|22.8|31.5|
> |Cross-Entropy (Eq. (4))|No|22.7|31.6|
> |None|Yes|22.3|30.8|
> |KL-Divergence|Yes|22.9|31.8|
> |**Cross-Entropy (Eq. (4))**|**Yes**|**23.0**|**32.0**|
>
> The above table provides an ablation study on 1% of COCO labels to assess the key elements in SoftER Teacher. ***SoftER Teacher  (last row) improves on both precision and recall over Soft Teacher (first row) via our proposed Entropy Regression module (Eqs. (4) and (5)) for proposal learning with complex affine transforms.***
>
> ### 5. The novel performance improvement is minor in the FSOD setting.
> Please see the above **2. General Response** on novel class performance for a detailed answer to your concern. In short, we argue that SoftER Teacher performs exceedingly well on novel classes with less parameters and labels by leveraging supplementary unlabeled data, when compared to TFA and Retentive R-CNN baselines adopting the same base FRCN architecture.
>
> ### 6. The authors do not show any numbers related to the training resources (memory and time).
> Again due to page limit, we put the Implementation Details in Appendix C of the supplementary material, called out in the main paper on L271. In short, we train on 8x A6000 GPUs each with 48GB of memory. One experiment takes between 12 hours and 10 days to complete. Please refer to Appendix C and the included source code for details on reproducibility. We will add details on training resources in the camera-ready version using the extra page.
>
> ### 7. SoftER Teacher adopts FRCN, how about the performance when replacing FRCN with other FSOD methods like DeFRCN?
> The goal of this work is to explore and analyze the contribution of unlabeled data for semi-supervised FSOD. However, we recognize that SoftER Teacher has room for improvement. We observe complementary properties of DeFRCN, DCFS, and Retentive R-CNN which, in principle, could be combined with SoftER Teacher to further advance FSOD without base degradation. Such in-depth investigation is better reserved for future work as it is beyond the scope of this paper.

---

> > ### Comment · Reviewer_F1pt · 2023-08-18
> >
> > The author seems to haven't fully explained my questions, $e.g.$,
> > 1. The proposed task of semi-supervised few-shot objects is similar to semi-supervised object detection. Since both base and novel classes are scarce, what's the meaning of splitting classes into base and novel? I can't see any practical significance in this task.
> > 2. FSOD assumes novel classes to be authentically rare, which means we can't find samples of novel classes, whether labeled or unlabeled, it seems semi-supervised learning conflicts with the definition of FSOD.
> >
> > Please carefully clarify my questions, I would keep my initial rating for now.

---

> > > ### Author Response · Authors · 2023-08-18
> > > **Thank you for replying to our rebuttal.**
> > >
> > > We thank Reviewer `F1pt` for replying to our rebuttal. We believe to have addressed your questions and concerns in our rebuttal, but the answers may have been buried in our responses. Please find below our additional clarifications to your concerns.
> > >
> > > **Q1. The proposed task of semi-supervised few-shot objects is similar to semi-supervised object detection. Since both base and novel classes are scarce, what's the meaning of splitting classes into base and novel? I can't see any practical significance in this task.**
> > >
> > > We explained in our rebuttal that the base dataset in FSOD benchmarks is assumed to be fully annotated with bounding boxes covering all instances of interest, whereas the novel set is sparsely, and randomly, labeled at $k$-shots, or bounding boxes, per class, with the potential for missing labels since multiple objects of the same class may appear in an image. In our approach, we simulate the scarcity of base labels by randomly sampling small fractions at {1,5,10} percent from the full base dataset, but the sampled fractions are still fully annotated with all instances having bounding boxes.
> > >
> > > Thus, ***we explained in our rebuttal that the benefit of the two-stage approach, and its practical significance, is to separate fully annotated base examples from sparsely annotated novel examples, thereby allowing for algorithms to optimize on both base and novel categories.*** To accomplish this goal, we introduce unlabeled data and our SoftER Teacher model in the two-stage procedure (Sections 3.2 and 3.3). For base pre-training, SoftER Teacher vastly expands base AP on both VOC (80.8 to 85.9) and COCO (39.3 to 44.4). For few-shot fine-tuning, SoftER Teacher improves on the strong Retentive R-CNN baseline by up to $+1.6$ novel AP on COCO (Table 1) and $+7.3$ novel AP on VOC (Table 2) while mitigating base forgetting to less than 9%.
> > >
> > > If we were not splitting base and novel classes into two stages, but train both under a single stage SSOD protocol, we would run into the issue of foreground objects being rejected as background, due to missing labels, resulting in a drastic reduction in base AP, as shown in our rebuttal.
> > >
> > > **Q2. FSOD assumes novel classes to be authentically rare, which means we can't find samples of novel classes, whether labeled or unlabeled, it seems semi-supervised learning conflicts with the definition of FSOD.**
> > >
> > > **We agree with the reviewer that one property of the novel class can be attributed to its rarity, *but the novel class is not necessarily only rare or long-tailed.*** By definition, the novel class is a new object category that the model has not yet learned. In practical scenarios, when we want to adapt the base detector to expand its base vocabulary to include a novel concept, a natural way is to find images containing such examples and annotate them. Or in our case, don't annotate the additional images at all, but rather use them as unlabeled data. The novel class, depending on object type, can occur at any distribution, from *frequent* to *common* or even *rare*. Thus, we believe our approach of leveraging unlabeled sources such as COCO-20 and COCO-unlabeled2017 is a fair and reasonable comparison to existing FSOD works since we do not assume large quantities of novel classes in the unlabeled set. **Our approach overcomes a fundamental limitation of prior works like LVC and MINI, in which they make an unrealistic assumption that novel classes necessarily be present in large amounts in the *base training dataset* to achieve robust performance on FSOD benchmarks.** By stark contrast, we do not use the base dataset as unlabeled data.
> > >
> > > Please see the above [1. General Response on the source of unlabeled data](https://openreview.net/forum?id=THDGuhN7LA&noteId=8h5dO1DL96) for additional experiments to help address your concern. We conducted two experiments on VOC0712 to measure the effectiveness of our approach by using unlabeled data "in the wild" containing many objects outside of the target domain. The first experiment uses the broader COCO-train2017 as unlabeled data, instead of COCO-20, in which the proportion of novel classes is low at roughly 4.6%. In the second experiment, we remove $16496$ images from COCO-train2017 that contain any novel instances. To be clear, our model does not see any instances of the novel classes, in both labeled and unlabeled sets, except during few-shot fine-tuning.
> > >
> > > ***We observe our SoftER Teacher model to be robust against strong domain mismatch between COCO and VOC datasets. Our approach does surprisingly well in the general scenarios where the novel classes are rare and completely absent in both labeled and unlabeled sets.*** We acknowledge that this observation is limited to experiments on VOC. We believe further experimentation and analysis are needed to determine if the trend holds on the more challenging COCO and LVIS datasets, using open-domain unlabeled sources like Objects365 and OpenImages, which is beyond the scope of this work.

---

### Official Review · Reviewer_Qe4L · 2023-07-09

**Soundness:** 2 fair
**Presentation:** 2 fair
**Contribution:** 2 fair
**Rating:** 4
**Confidence:** 5

**Summary:**

This paper focuses on Semi-Supervised Few-Shot Object Detection, where both base classes and novel classes have few labeled training set, along with abundant unlabeled data. For model architecture, the softer teacher is proposed to train with unlabeled data and a teacher-student framework. Experiments demonstrate strong performance using only 10% base labels.

**Strengths:**

The idea of introducing unlabeled data for few-shot object detection is interesting and has great value for real-world application. The idea of  reducing the number of labeled data for base classes in few-shot object detection is also interesting.

**Weaknesses:**

My major concerns are as below:

1. What is the sources of unlabeled images? Do the unlabeled images have both base-class and novel-class instances? In this way, it is no surprise that adding abundant images could improve the proposal recall and detection results of few-shot novel classes.

2. The training framework in figure 2 and Section 3.3 are not exactly the same. In Figure 2, the unlabeled images are used for both base-class pre-training and few-shot fine-tuning. But in Section 3.3, it seems that the second stage of few-shot fine-tuning do not use additional unlabeled image. Clarifications are need. If the second stage also use unlabeled images, can we merge the second stage into the first stage because both base classes and novel classes are few-shot? We do not need to have two stages for training in that case, and the problem becomes semi-supervised object detection and each class has very few labeled images. Thus, what is the difference between traditional semi-supervised object detection?

3. This work has improved overall performance of base and novel class. Although the performance of novel class improves compared some baseline model (e.g., Faster RCNN), but has far worse performance compared to the SOTA [1,2]. Does this mean that the additional unlabeled images only work well with base classes, but not for novel class? Using unlabeled image is perhaps a right way for semi-supervised object detection. But does using unlabeled images is the right way for few-shot object detection?

[1] Qiao, Limeng, Yuxuan Zhao, Zhiyuan Li, Xi Qiu, Jianan Wu, and Chi Zhang. "Defrcn: Decoupled faster r-cnn for few-shot object detection." In Proceedings of the IEEE/CVF International Conference on Computer Vision, pp. 8681-8690. 2021.

[2] Kaul, Prannay, Weidi Xie, and Andrew Zisserman. "Label, verify, correct: A simple few shot object detection method." In Proceedings of the IEEE/CVF conference on computer vision and pattern recognition, pp. 14237-14247. 2022.

4. The Table 1 and Table 2 are not complete. Table 1 lacks the comparison of Faster R-CNN (Our Impl.) and Soft Teacher (Our Impl.) as in Table 2. Table 2 lacks the comparison of the latest method for few-shot object detection (e.g. LVC, DeFRCN) as in Table 1.

5. The Figure 3 (b) and (c) are very confusing. I can only find one red box. Does it mean that vanilla FRCN-base only have one proposal? This is weird.

6. What is the difference between soft teacher and softer teacher? In L201-L207, the author mentions that soft teacher has an aggressive threshold of 0.9 which is not good. How did the authors address this problem? I do not find the answer in the main text. L208-L225 seems to be a simple extension of soft teacher without big changes.




==========================================================================================

After reading author's rebuttal and other reviews, some of the concerns about technical details are clear. But the major concern about the significance of doing few-shot base/novel partition is still there. I would suggest the authors make the problem setting simpler to get broader impact.

**Questions:**

Please see the weaknesses above

**Limitations:**

Please see the weaknesses above

---

> ### Author Rebuttal · Authors · 2023-08-09
>
> ### We thank Reviewer `Qe4L` for your positive and constructive feedback. Please find below our responses, along with additional experiments, to address your questions and concerns.
> ### 1. What is the source of unlabeled images? Do the unlabeled images have both base and novel classes? It is no surprise that adding abundant images could improve proposal recall and FSOD results.
> Please refer to the above **1. General Response** on the source of unlabeled data for a detailed answer to your question. In short, we don't make a strong assumption that the unlabeled images necessarily contain abundant novel instances, unlike the prior work of LVC [CVPR22] and MINI [aXiv22].
>
> Although "no surprise" in hindsight, there was little prior analysis or empirical study on *how or why* unlabeled data could improve proposal recall in the FSOD setting. We contribute an insightful empirical analysis in Sections 3.1 and 4.2 linking the role of unlabeled data to FSOD by way of proposal recall. Moreover, Section 4.2 explains that ***while the strong Soft Teacher [ICCV21] baseline can harness unlabeled data for FSOD, SoftER Teacher demonstrates superior learning by further boosting object recall with our entropy regression module.*** For semi-supervised FSOD (Table 3), SoftER Teacher is the right model for the task to improve on Soft Teacher by +1.7 base class AR, which yields a gain of up to +1.5 novel class AP. We argue that it is a combination of unlabeled data and our algorithmic contribution in SoftER Teacher that works well together to boost proposal recall and FSOD results.
>
> ### 2. The training framework in Figure 2 and Section 3.3 are not exactly the same.
> We argue that the schematic in Figure 2 closely follows the text in Section 3.3. L245 explicitly states that we use unlabeled data in the fine-tuning phase, with the rest of the text describing the procedural details.
>
> ### 3. Can we merge the second stage into the first stage because both base and novel classes are few-shot, and the problem becomes SSOD?
> To properly answer this question and illustrate the impact of our approach, we train SoftER Teacher on few-shot examples of base and novel classes, supplemented with COCO unlabeled2017 images, following the SSOD formulation described in Section 3.2. Denoted in the table below as "One-Stage Semi-Supervised", this setting exhibits drastic reduction in base performance at the trade-off in slightly better novel detection compared to our proposed two-stage approach, and hence is impractical in real-world scenarios since samples at test time may contain instances of both base and novel classes.
>
> |Training Protocol|10-Shot bAP|30-Shot bAP|10-Shot nAP|30-Shot nAP|
> |-|-|-|-|-|
> |One-Stage Semi-Supervised|11.0|16.9|11.6|15.3|
> |Proposed Two-Stage Few-Shot (Section 3)|37.2|38.6|10.6|12.3|
>
> Recall that few-shot instances are sampled at the bounding box level. So a novel image containing 3 cats may not be fully annotated, but only one box (1-shot) is annotated while the other two ignored. Thus, FSOD is very different from SSOD, where a small fraction of images is fully labeled with boxes covering all instances of interest. As shown in the above results, if we approach the FSOD problem as SSOD, we would run into the issue of foreground objects being rejected as background (i.e., missing labels), along with extreme label scarcity that would be difficult to rectify with only unlabeled data.
>
> ### 4. The performance of novel class is worse compared to SOTA.
> Please see the above **2. General Response** on novel performance for a detailed response to your concern.
>
> ### 5. Do the additional unlabeled images only work well with base classes, but not for novel classes?
> Section 3 describes in-depth how unlabeled images can contribute significant improvements on both base and novel classes. For base pre-training, unlabeled images vastly expand base AP on both VOC (80.8 $\rightarrow$ 85.9) and COCO (39.3 $\rightarrow$ 44.4). For few-shot fine-tuning, Figure 6 in Appendix A.2 shows that unlabeled images contribute up to +3.3 novel class AP, while mitigating base forgetting to less than 9% (Table 5 in Appendix A.3).
>
> ### 6. Tables 1 and 2 are not complete.
> Due to page limit, we could include only the essential comparisons in Tables 1 and 2. We will include the additional comparisons, per your suggestion, in the camera-ready version with the extra page.
>
> ### 7. Figures 3b and 3c are very confusing. I can only find one red box. Does it mean that vanilla FRCN-base only has one proposal? This is weird.
> Yes, Figures 3b and 3c show the vanilla FRCN-base fails to capture novel foreground objects in low-label regimes, with only 1 red box appearing in the 10% base label setting. These proposals have confidence scores $> 0.9$. The lack of high-quality proposals produced by FRCN-Base further demonstrates the utility of unlabeled for FSOD and validates the motivation for our SoftER Teacher approach.
>
> ### 8. What is the difference between Soft Teacher and SoftER Teacher? It seems to be a simple extension of Soft Teacher without big changes.
> L234-243 in Section 3.2 argue that our SoftER Teacher contributes a non-trivial extension to Soft Teacher with our Entropy Regression (ER) module for proposal learning with complex affine transformations, which has not been attempted before. Soft Teacher uses an aggressive threshold of 0.9, resulting in overall poor recall. ER addresses this key weakness to enhance proposal recall by allowing SoftER Teacher to tap into abundant region proposals to learn diverse representations across scale, color, and geometric perturbations, the results of which translate to convincingly better FSOD performance (Figure 5 and Table 3).
>
> We arrived at our SoftER Teacher model based on meticulous research and validated design choices grounded on insightful empirical findings. As such, we believe SoftER Teacher is a good technical contribution to the community as it is well suited for both semi-supervised and few-shot tasks.

---

### Official Review · Reviewer_2E7Y · 2023-07-10

**Soundness:** 3 good
**Presentation:** 3 good
**Contribution:** 3 good
**Rating:** 5
**Confidence:** 4

**Summary:**

This article has done a meaningful work, which is a object detection method that combines few-shot with semi-supervised learning. The author introduces a SoftER Teacher for semi-supervised object detection in few-shot scenarios. SoftER Teacher enhances the quality of region proposals to substantially boost semi-supervised FSOD. Compared with LVC, DeFRCN and other methods, the performance has been improved.

**Strengths:**

- This task is very meaningful. As far as I know, the traditional few-shot object detection task is difficult to be directly applied to the industry, and the method combined with semi-supervised target detection will be a good solution. (Although the method in this paper is not the first to consider the combination of few-shot and semi-supervised.)
- It is good to see that the author provides the source code in the supplementary material, which provides a guarantee for the reproducibility of this article.
- The authors present rich experimental results in the article and supplementary material.

**Weaknesses:**

- To my acknowledgment, the current mainstream FSOD method is verified on MS COCO 2014, not MS COCO 2017. "Consistent with the current literature on FSOD" may be ambiguous.
- In Table 1, since the author did not report the results of LVC in 5-Shot, the experimental results of LVC are from the original article? But considering that our method has unlabeled data for additional testing, it seems unfair to compare the results with the LVC experimental setting.
- In addition to the comparison with the FSOD method, it would be better to add some comparisons with the SSOD method (under the Few-shot setting).
- In Table 1, I found that the results of the novel class seem to be relatively weak, what is the reason? Because the method in this paper utilizes additional data.

**Questions:**

My main concern is the experimental comparison, such as whether the comparison method has changed the experimental setting. Another concern is the performance of the novel class. If the author can convincingly solve my problem, I would like to improve my score.


I also suggest that the author can also refer to a method on arxiv [1], which seems to have the same task as this article (Of course, this article does not need to compare with its method at this stage.).

[1] Cao, Yuhang, et al. "MINI: mining implicit novel instances for few-shot object detection." *arXiv preprint arXiv:2205.03381* (2022).

**Limitations:**

The authors describe the limitations of this paper in the supplementary material.

---

> ### Author Rebuttal · Authors · 2023-08-09
>
> ### We thank Reviewer `2E7Y` for your positive and constructive feedback. Please find below our responses, along with additional experiments, to address your questions and concerns.
> ### 1. The current mainstream FSOD method is verified on COCO 2014, not COCO 2017.
> The reviewer is correct in that the current established FSOD evaluation protocol is on COCO 2014, not COCO 2017. However, both COCO 2014 and 2017 share the same images. The only difference between the two is the number of validation images (41k images for COCO val2014 and 5k for COCO val2017). The authors of TFA [ICML20] created the original FSOD benchmark by sampling from COCO 2014 a random subset of 5k images for validation and used the rest in the training split. Thus, both train/val splits from COCO 2014 and 2017 should effectively be the same, with minor variance due to the sampling process. Our preliminary experiments on both COCO 2014 (following the TFA splits) and the official COCO 2017 splits verified that the difference is indeed minor up to some statistical noise (see table below). The benefit of using the official COCO 2017 splits is to remove the dependency on the random train/val splits created by the TFA benchmark and to maintain consistency with our proposed semi-supervised FSOD benchmark in Table 3.
>
> |Model|Dataset|5-Shot bAP|10-Shot bAP|30-Shot bAP|5-Shot nAP|10-Shot nAP|30-Shot nAP|
> |-|-|-|-|-|-|-|-|
> |FRCN|COCO 2014|36.0 $\pm$ 0.3| 36.1 $\pm$ 0.1|37.2 $\pm$ 0.1|3.8 $\pm$ 0.7|6.2 $\pm$ 0.6|9.3 $\pm$ 0.6|
> |FRCN|COCO 2017|36.0 $\pm$ 0.2|36.0 $\pm$ 0.2|37.0 $\pm$ 0.2|3.7 $\pm$ 0.4|6.1 $\pm$ 0.3|9.6 $\pm$ 0.2|
> |SoftER Teacher|COCO 2014|41.8 $\pm$ 0.1|41.8 $\pm$ 0.4|42.6 $\pm$ 0.4|7.0 $\pm$ 0.3|9.6 $\pm$ 0.4|12.4 $\pm$ 0.5|
> |SoftER Teacher|COCO 2017|41.8 $\pm$ 0.2|41.9 $\pm$ 0.2|42.7 $\pm$ 0.1|7.5 $\pm$ 0.4|10.0 $\pm$ 0.4|12.5 $\pm$ 0.5|
>
> ### 2. In Table 1, are the experimental results of LVC from the original article? It seems unfair to compare the results with the LVC experimental setting with additional unlabeled data.
> The LVC [CVPR22] method did not report results for the 5-shot setting. All results in Table 1 are reported from the respective original works. To our knowledge, we believe that LVC only performed single sample runs in their few-shot experiments (hence the lack of error bars), instead of following the established protocol of repeated sample runs over multiple random seeds. As such, their results may have been over-estimated due to the high variance of few-shot training samples. The previous works of TFA, Retentive R-CNN, DeFRCN, and DCFS all have reported marked reduction in novel performances with 10 repeated sample runs compared to a single sample run. It is unclear if the strong novel performances of LVC hold in the same repeated setting.
>
> We believe our work is a fair head-to-head comparison with LVC because we use the same amount of labeled training data as LVC and other methods. The only difference in our work is the addition of unlabeled data, which is allowed in the comparison because unlabeled images are not an automatic guarantee for improved performance. Thus, the comparison with LVC brings out two advantages attributed to our approach: (a) unlike LVC, we do not assume abundant novel classes exist in the base training set, and we were conscientious to not include the base dataset as a source of "unlabeled" images; and (b) our approach exhibits less than 9% in base forgetting compared to 19% for LVC. It is also unclear if LVC can achieve strong FSOD performance assuming only 10% of base labels (vs. 100%) are available instead.
>
> ### 3. In addition to the comparison with the FSOD method, it would be better to add some comparisons with the SSOD method under the few-shot setting.
> Our work generalizes two SSOD models to the few-shot setting: Soft Teacher [ICCV21] and our proposed SoftER Teacher. L308 states that while the strong Soft Teacher baseline can harness unlabeled data for semi-supervised FSOD, SoftER Teacher demonstrates superior learning by further boosting object recall in Soft Teacher. For semi-supervised FSOD (Table 3), SoftER Teacher improves on Soft Teacher by +1.7 base class AR, which yields a gain of up to +1.5 novel class AP.
>
> Moreover, Figure 5 presents new empirical insight into why SoftER Teacher is a better few-shot detector by analyzing semi-supervised FSOD as a function of proposal quality. SoftER Teacher produces better proposal recall than Soft Teacher, which translates to convincingly stronger semi-supervised FSOD. Future work would examine if our empirical finding can be extended to a more general case with other SSOD formulations including one-stage detectors.
>
> ### 4. In Table 1, I found the results of the novel class seem to be relatively weak, what is the reason?
> Please see the above **2. General Response** on novel class performance for a detailed answer to your question. In short, we argue that SoftER Teacher performs exceedingly well on novel class performance while using less parameters and labels than the comparable TFA and Retentive R-CNN baselines adopting the same base FRCN architecture.
>
> ### 5. I also suggest that the author can also refer to MINI, which seems to have the same task as this article.
> From our understanding, MINI is similar to LVC in that *both methods mine novel targets as auxiliary samples from the base training set*, thereby making a strong and explicit assumption that novel instances must necessarily be present in the training set. With this assumption, MINI achieves impressive performances on FSOD benchmarks. However, this assumption is unrealistic in real-world few-shot settings because novel objects may not exist in the base dataset in large quantities. Moreover, it seems that the training protocol is overly complex, introducing four additional hyper-parameters, making MINI impractical in real-world applications. We will include this reference in the camera-ready version, along with a discussion comparing it to our method.

---

> > ### Comment · Reviewer_2E7Y · 2023-08-12
> > **Reply to the author's rebuttal**
> >
> > I am very grateful to the author for carefully answering my questions, I am very grateful. I think the author of the details about the COCO dataset should carefully add it to the text for more convenient reference in subsequent articles. Since the author addressed my concerns, I decided to raise the score to **borderline accept**.

---

> > > ### Author Response · Authors · 2023-08-13
> > > **Thank you for your support on our work.**
> > >
> > > Dear Reviewer `2E7Y`,
> > >
> > > We would like to sincerely thank you for contributing your time to serve as a reviewer for NeurIPS 2023. And we are grateful to you for providing your constructive feedback, replying to our rebuttal, and raising your score.
> > >
> > > Per your suggestion, we will include in the revised paper the details about COCO 2014 vs. 2017 to help avoid potential confusion and ambiguities for the readers.
> > >
> > > May we ask why you think our paper merits a Borderline Accept? According to the description, Borderline Accept means that you still have some concerns about the paper, e.g., limited evaluation. Have we adequately addressed all of your concerns and questions in our rebuttal? If you have additional concerns, we are happy to further discuss and help address them.
> > >
> > > Given your strong assessments on the quality of the paper in terms of **3 - Good Soundness**, **3 - Good Presentation**, and **3 - Good Contribution**, along with **your positive comments on the meaningfulness of our work and how our approach is a good solution for realistic few-shot settings in practical applications**, we would be very grateful if you could offer your clear and enthusiastic support for our work beyond the Borderline Accept rating, as a reflection of your overall positive review of our paper.

---

### Official Review · Reviewer_4p6Z · 2023-07-26

**Soundness:** 4 excellent
**Presentation:** 4 excellent
**Contribution:** 4 excellent
**Rating:** 6
**Confidence:** 4

**Summary:**

The paper approaches the task of Few-shot Object Detection (FSOD) from a semi-supervised perspective, where in addition to base classes data it uses unlabeled data during the base-pretraining phase, and then fine-tunes on the combination of base and available novel data using the best design choice of freezing appropriate layers (backbone, FPN and RPN) following the past works. The benefit of the approach comes with a higher bar on fully-supervised base class performance, which is attributed to training on additional unlabeled data. This higher performance then translates to a better overall (base + novel) classes performance in the fine-tuning phase, and establishes the effectiveness of semi-supervised learning on the FSOD task. The authors propose a SoftER Teacher approach which, in addition to the Soft Teacher loss, adds a consistency loss between teacher and student using at proposal-level. The authors show that this leads to better proposals (using recall) in low-data regimes.

**Strengths:**

- The paper is well-written with extensive experiments and ablations, and the semi-supervised exposition in the FSOD setup is much appreciated with a potential for realistic low-shot setups
- The paper is well-positioned with respect to prior related works


**Weaknesses:**


- Table 2 on VOC07 and lower performance of Novel classes in low-shot setup
    - Retentive R-CNN compared to SoftER exhibits a trend that its 1-shot performance is much higher that the proposed SoftER, and this trend gets reversed with more shots (such as 10-shot)
    - The authors acknowledges the phenomenon in lines 295-296, and in line 297 mentions that Retentive R-CNN “generally falls behind on novel class performance”. However, this isn’t true in low-data setup (1-shot).
    - This trend, however, doesn’t appear in the COCO dataset where novel class performance in 1-shot case is low for both Retentive R-CNN and the proposed SoftER approach
    - My question is:
        - Do the authors have any intuition for this behaviour?
        - The authors leverage COCO-20 and COCO unlabeled2017 as he unlabeled data source for VOC (Table 2) and COCO (Table 1) experiments. Do the authors think that the domain mismatch between VOC and COCO is reflected in low-shot novel class performance in the case of VOC (Table 2)?
        - In general, the proposed approach does better with relatively higher-shot regimes, which also appears in the claims made in the paper - such as Fig 1 (30-shot). But as a reader, there seems to be little explanation about low-shot regimes, which runs counter-intuitive since the approach uses additional unlabeled data compared to prior approaches, and is expected to perform well especially in low-data regimes

- Presence of novel classes in unlabeled data
    - Line 277-278 mentions the use of COCO-20 for VOC and COCO unlabeled2017 as the source of unlabeled data for VOC and COCO respectively
    - This makes the assumption that novel classes in the plots of VOC and COCO experiments are necessarily present in the unlabeled set
    - In general scenarios, such assumption may not hold true. Do the authors have some intuition of how the proposed approach would work if the percentage of novel classes in the chosen unlabeled set is low / absent?


### Minor concerns
- Figure 3a for more percentage of base labels
    - Not a head-to-head comparison between FRCN-Base and FRCN-Base + Unlabeled, since the latter assumes more data
    - Does the difference narrow with more percentage of base labels?

- Conflicting claims
    - Fig 1: exhibiting less than 7% in base degradation
    - line 54: exhibiting less than 9% in base forgetting


### Justification of the rating
My main concern is highlighted above. In general, explanations towards the above mentioned concern would help the readers


**Questions:**

Put together in the weaknesses section

**Limitations:**

Yes

---

> ### Author Rebuttal · Authors · 2023-08-09
>
> ### We thank Reviewer `4p6Z` for your positive, insightful, and constructive feedback. Please find below our responses, along with additional experiments, to address your questions and concerns.
> ### 1. Table 2 on VOC07 - low novel class performance in 1-shot case.
> Table 2 compares the few-shot results of our models based on the *ResNet-50 backbone* to competing methods based on the *ResNet-101 backbone*. ***It is remarkable that SoftER Teacher with ResNet-50 vastly expands the supervised base AP from 80.8 to 85.9, and incurs negligible base forgetting of less than 1.6%, while exceeding MPSR [ECCV20], TFA [ICML20], and Retentive R-CNN [CVPR21] with ResNet-101 by a notable margin on most metrics.***
>
> To our knowledge, we believe that both MPSR and Retentive R-CNN only performed single sample runs in their VOC few-shot experiments (hence the lack of error bars), instead of following the established protocol of repeated sample runs over multiple random seeds. As such, their results may have been over-estimated due to the high variance of few-shot training samples, as originally reported by TFA and exhibited by large 95% confidence intervals in Table 2. For example, ***our {min, max} novel AP from 10 repeated runs for the 1-shot setting is {21.5%, 41.9%}, respectively, the max of which is on par with MPSR and Retentive R-CNN***. In general, the previous works of TFA, DeFRCN, and DCFS all have reported marked reduction in novel performances with repeated sample runs when compared to a single sample run. We believe that if MPSR and Retentive R-CNN were to perform repeated sample runs over 10 random seeds, and if our approach was based on the ResNet-101 backbone, then the observed performance gap for the novel AP in the 1-shot setting would not exist. The reason why we experimented with ResNet-50 is to demonstrate parameter-efficient learning with SoftER Teacher and to see how far unlabeled data takes us with a smaller backbone architecture.
>
> ### 2. Presence of novel classes in unlabeled data.
> Thank you for bringing up this insightful question. Please refer to the above **1. General Response** on the source of unlabeled data for additional experiments to address your concern. In short, we observe SoftER Teacher to be robust against domain mismatch between COCO and VOC datasets. Our approach does surprisingly well in two scenarios where the percentage of novel classes in the chosen COCO-train2017 "unlabeled" set is (a) low at roughly 4.6% and (b) completely absent, with the best case scenario having an unlabeled set containing targeted base + novel classes (e.g., COCO-20). We acknowledge that this observation is limited to few-shot experiments on VOC; we believe further exploration, experimentation, and analysis are needed to determine if the trend holds on the more challenging COCO and LVIS datasets using large-scale, open-domain unlabeled datasets like Objects365 and OpenImages, which is beyond the scope of this work. We believe our approach to FSOD overcomes the fundamental limitation of prior works in real-world scenarios by not assuming the presence of large amounts of base and novel instances in either labeled or unlabeled dataset.
>
> ### 3. Figure 3a - Not a head-to-head comparison between FRCN-Base and FRCN-Base + Unlabeled, since the latter assumes more data. Does the difference narrow with more percentage of base labels?
> The comparison in Figure 3a between the supervised FRCN-Base model and semi-supervised FRCN-Base + Unlabeled model is a standard, well-established protocol routinely employed in the semi-supervised learning literature to measure the effectiveness of SSL algorithms. We argue that Figure 3a is a fair comparison since both models use the same amount of labeled training data. The addition of unlabeled images in the semi-supervised pipeline is allowed in the comparison because unlabeled images are not an automatic guarantee for improved performance. We present Figure 3a to illustrate the contribution of unlabeled data to boost proposal recall of novel categories for the better discovery of novel classes during few-shot fine-tuning, and to motivate the design and development of SoftER Teacher as a well-suited model to address the unique task of semi-supervised few-shot detection at low-label regimes.
>
> The performance gap between FRCN-Base and FRCN-Base + Unlabeled becomes narrow with more percentage of base labels, as shown by additional experiments in the table below, using the standard metric AR@300 for quantifying proposal recall of both base + novel categories. Interestingly, with 100% of base labels, the difference in proposal recall between the two models is immaterial with FRCN-Base + Unlabeled edging out the FRCN-Base model by +0.38 point. This result suggests that the addition of unlabeled data during base pre-training can help boost base representation learning *and also* proposal recall, especially at low-label regimes, the *combination* of which should lead to both better transferability and discovery of novel classes in the subsequent fine-tuning phase. We observe supporting experimental evidence in Table 2 in the main paper and Table 10 in Appendix B.4, where the fully supervised FRCN trails behind SoftER Teacher on both base and novel performances, even though proposal recall between the two is effectively the same.
>
> | % Base Labels | 1% | 5% | 10% | 100% |
> |-|-|-|-|-|
> | FRCN-Base | 21.91 | 28.94 | 30.97 | 39.92 |
> | FRCN-Base + Unlabeled | 33.63 | 35.49 | 36.83 | 40.30 |
> | Difference | +11.72 | +6.55 | +5.86 | +0.38 |
>
> ### 4. Conflicting claims.
> Thank you for catching this typo. We will fix the caption in Figure 1 to say "less than 9% in base forgetting" to match the text throughout the paper.

---

### Author Rebuttal · Authors · 2023-08-09

### We sincerely thank all reviewers for their thoughtful and constructive feedback. We would like to address two concerns common in the reviews.
### 1. General Response - What is the source of unlabeled data? [`4p6Z`, `Qe4L`]
Per L261, we leverage COCO-20 and COCO-unlabeled2017 as unlabeled data for VOC and COCO few-shot experiments, respectively. COCO-20 contains images with VOC base and novel instances, along with other objects outside of VOC domain. And COCO-unlabeled2017 has unknown base-novel class distribution, along with other objects outside of COCO 80 classes. To our knowledge, ***we are the first to use external supplementary unlabeled images for FSOD***, especially COCO-20 which exhibits strong domain mismatch with VOC. By contrast, the previous work of LVC [CVPR22] and MINI [aXiv22] make an explicit assumption that abundant novel instances must necessarily be present in the ***base training set***, which is unrealistic and a fundamental limitation of LVC and MINI in real-world applications.

We do not make a strong assumption that novel classes must exist in large quantities in unlabeled images. In practical scenarios, it is natural to collect unlabeled data having both base and novel instances. For example, when one wants to further detect a novel object that is not in the base categories, a reasonable way is to find additional targeted images containing such objects and annotate them. Or in our case, don't annotate the additional images at all, but rather use them as unlabeled data. Thus, a unique benefit of our FSOD approach reduces the human burden on annotating a lot of required images, as shown in Table 3, which is in stark contrast to others requiring an abundance of base labels for robust FSOD.

We agree with Reviewers `4p6Z` and `Qe4L` by recognizing that the choice of unlabeled data can be difficult in general scenarios where strong domain mismatch can occur. To address this concern, we perform two few-shot experiments on VOC0712 to demonstrate the effectiveness of our approach by leveraging unlabeled data "in the wild" containing many objects outside of the target domain. The first experiment uses the broader COCO-train2017 as unlabeled data, instead of COCO-20, in which the proportion of novel classes is low at roughly 4.6%. In the second experiment, we filter out all images from COCO-train2017 that contain at least one instance of the novel class, thereby removing the assumption that novel instances must be present in the unlabeled set.

|Model|Unlabeled|1-Shot bAP|5-Shot bAP|10-Shot bAP|1-Shot nAP|5-Shot nAP|10-Shot nAP|
|-|-|-|-|-|-|-|-|
|FRCN|None|81.8|82.3|82.2|36.2|53.3|58.7|
|SoftER Teacher|COCO-20|84.5|85.2|85.5|38.6|57.8|63.4|
|SoftER Teacher|COCO-train2017|83.4|84.4|84.4|38.4|57.4|63.4|
|SoftER Teacher|COCO-train2017-no-novel|82.7|83.4|84.0|36.8|56.8|62.8|

Recall our empirical analysis in Sections 3.1 and 4.2 connects the role of unlabeled data to FSOD by way of proposal recall. We show that unlabeled data can help boost proposal recall on novel categories, which should lead to better discovery of novel classes during fine-tuning. Intuitively, we expect the best FSOD performance if the unlabeled images contain targeted base + novel classes (COCO-20). This is reflected in the above results. Surprisingly though, we also observe strong robustness of our approach in the general scenarios where the percentage of novel classes in the chosen unlabeled set is low (COCO-train2017) or completely absent (COCO-train2017-no-novel).

Future work would explore in depth if the observed trend with VOC also holds for more challenging datasets such as COCO and LVIS, using open-domain unlabeled data sources like Objects365 and OpenImages.

### 2. General Response - On weak novel class performance. [`2E7Y`, `Qe4L`, `F1pt`]
The goal of this work is to explore and analyze the contribution of unlabeled data for semi-supervised FSOD. As such, we adopt the vanilla FRCN as our base detector, transform it into SoftER Teacher with unsupervised losses, and add unlabeled data. We keep everything else about the base architecture the same, including the backbone, FPN, RPN, and RoI heads, to avoid confounding factors due to model design and training protocol. Thus, if we directly compare our SoftER Teacher to TFA [ICML20] and Retentive R-CNN [CVPR21], which is a reasonable and fair comparison since they all use the same base FRCN model and train on the same amount of labeled images, then ***our approach surpasses both TFA and Retentive R-CNN on most novel class settings in Tables 1 and 2 while being parameter-efficient with a smaller ResNet-50 backbone.***

Most notably, Table 3 shows that SoftER Teacher surpasses the novel class performance of Retentive R-CNN on all shots under consideration while requiring only 10% of base labels, further demonstrating its effectiveness. Therefore, ***we argue that SoftER Teacher can do more on novel class performance with less parameters and labels, which should be both a technical contribution and insightful empirical finding of interest to the community, the basis of which could inspire future research.***

Reviewers `2E7Y`, `Qe4L`, and `F1pt` mentioned that SoftER Teacher exhibits relatively poor novel class performance when compared to recent SOTA methods like LVC, DeFRCN, and DCFS. However, it is important to point out that these methods make unrealistic assumptions about the training dataset and/or heavily modify the underlying base architecture in such a way that promotes strong novel performance. These methods also exhibit significant base forgetting (11%–DCFS, 17%–DeFRCN, and 19%–LVC), which is an undesirable outcome since samples at test time may contain both base and novel objects. Lastly, these methods all achieve SOTA results at the requirement that 100% of abundant base labels must be available. It is unclear if these SOTA advances are competitive to SoftER Teacher on FSOD performance if only 10% of base labels are available instead.

---

### Decision · Program_Chairs · 2023-09-21

**Decision:**

Reject

**Comment:**

Paper received largely borderline ratings of: 1 x Weak Accept, 1 x Borderline Accept, and 2 x Borderline Reject. Importantly the confidence of the more negative reviewers was higher (5), compared to the more positive ones (4). A number of concerns were identified by the reviewers in the reviewing process. This included: (i) lower performance on novel classes in low-shot setup, (ii) unrealistic assumption of presence of novel classes in unlabeled data, (iii) issues with experimental setting, as well as (iv) some questions on the motivation. A through rebuttal was provided and addressed some of these concerns. In particular, the two positive reviewers are satisfied, while the more negative ones have not fully conceded their points in discussion post-rebuttal. Overall, paper was just at the casp of the borderline score, with no reviewer (including positive ones) strongly championing it. AC and SAC having read all the available materials (reviews, rebuttal, paper itself) do not see sufficient enough evidence that would push it above the acceptance threshold. As such, recommendation is to Reject the paper at this time.